# Knowledge Management and Total Quality Management Impact on Employee Effectiveness in Emerging Industries: Case of Tunisian Small and Medium Enterprises

**Fatma Lehyani [1], Alaeddine Zouari [1,2,*], Ahmed Ghorbel [3,4], Michel Tollenaere [5,6] and José Carlos Sá [7,8]**

1   OLID Laboratory LR19ES21: "Optimisation Logistique et Informatique Décisionnelle", University of Sfax, Technopole of Sfax, BP 1164, Sfax 3021, Tunisia
2   Higher Institute of Industrial Management, University of Sfax, Technopole of Sfax, BP 1164, Sfax 3021, Tunisia
3   Faculty of Economic and Management Sciences, University of Sfax, R Airport Km4, BP14, Sfax 3018, Tunisia
4   CODECI Laboratory, University of Sfax, R Airport Km4, BP14, Sfax 3018, Tunisia
5   Grenoble Institute of Technology, University of Grenoble-Alpes, 46 Av. Felix Viallet, 38031 Grenoble, France
6   G-SCOP Laboratory, University of Grenoble-Alpes, 46 Av. Felix Viallet, 38031 Grenoble, France
7   School of Engineering, Polytechnic of Porto (ISEP), 4200-072 Porto, Portugal
8   Institute of Science and Innovation in Mechanical and Industrial Engineering (INEGI), 4200-465 Porto, Portugal
*   Correspondence: ala.zouari@isgis.usf.tn

**Abstract:** Employee effectiveness is highly important for all economic activities. Several factors can affect its degree, either positively or negatively. In this vein, this work aims to examine the impact of knowledge management and total quality management on employee effectiveness in the industry of emerging countries. For that, Tunisian small and medium enterprises were taken as an example. The proposed methodology consists first of providing a research model linking the correlation between knowledge management elements, total quality management practices, and employee effectiveness. Then, a survey was designed and broadcast to more than 3000 Tunisian small and medium enterprises. Hence, 206 responses were collected from several industrial fields, and collected data analysis was achieved by SPSS software. For testing research hypotheses, multiple regression analysis, factor analysis, and structural equation modeling were employed. The finding points out that total quality management and knowledge management have a positive impact on staff effectiveness. This impact is highlighted through the roles of knowledge management elements and total quality management practices on human resources behavior and their ability enhancement. Consequently, a significant increase in productivity can be seen in the operational processes of the company. This work is one of the first studies to research total quality management and knowledge management impact on staff effectiveness in Tunisian small and medium enterprises. Besides, it reflects the maturity and the awareness of respondent companies' managers to the practice of these concepts in emerging economies.

**Keywords:** total quality management; knowledge management; employee effectiveness; structural equation modeling; empirical research

## 1. Introduction

The effective and efficient human factor is highly needed for economic organizations. The degree of employees' effectiveness is one of the most important factors in achieving a company's objectives. Thus, the workers' effectiveness and productivity are not inherent only to employees but also to the company's organizational structure and manager's will. Hence, managers must select compatible employees for the job requirements and commit to providing the main success factor to staff effectiveness, such as organizational culture and leadership [1], work environment and workload [2], employee involvement, and engagement [3], employee motivation, compensation, and reward [4], etc. Several

other catalysts can help reach these objectives. For example, knowledge sharing has a competitive advantage on digital start-ups, according to Tajpour et al. [5]. Besides, knowledge management (KM) has a strong impact on employee job satisfaction [6,7]. Likewise, KM will increase job performance [8,9]. In addition, KM positively impacts the employee's commitment [10] and employee service quality [11]. Thus, employee performance, which is the result of work in quality and quantity achieved by an employee in carrying out his duties in accordance with the responsibilities given to him, must continually be improved to increase employees' implication at work and create added value [12]. So, how do knowledge management (KM) and total quality management (TQM) affect staff effectiveness?

KM is part of a new management approach that aims for continuous improvement across all processes in the product and service value chain. The logic of knowledge management focuses on the knowledge management process and the motivation of employees to make work easier and more efficient [13]. This logic has various advantages for companies in terms of employee performance and efficiency [14]. According to Sadikoglu and Olcay [15], effective knowledge management ensures that employees get timely, reliable, consistent, accurate, and necessary data and information to perform their work effectively in the company. In this context, Tajpour et al. [16] claim that organizations should ensure access to knowledge, preserve intellectual property, and set standards for providing knowledge to employees. Furthermore, human capital theory suggests that organizations receive economic value from the knowledge, skills, and experience of their employees and that this capital can be increased through training and education [17]. Likewise, TQM affects all the product life cycle phases and requires the involvement of all employees in all departments. In this context, TQM can be defined as a quality management approach whose objective is to achieve a very broad mobilization and involvement of all staff to achieve perfect quality. Its successful implementation requires exemplary management and leadership to create the necessary ripple effect. According to Usrof and Elmorsey [18], human resource management and total quality management are strategically and tactically important to gain a competitive advantage through a variety of factors. These two management strategies have significant positive effects on the competitiveness of human resources [19]. According to Ghani et al. [20], the implementation of TQM practices is an ongoing process of employee engagement that encourages employees to make an effort to get involved and integrate to improve the effectiveness of their tasks and position in the organization. However, among the rare articles which have jointly treated the impact of KM and TQM on staff effectiveness, Saffar and Obeidat [21] have indicated that TQM practices have a significant impact on employee performance through knowledge sharing. Moreover, based on the results of Lehyani et al. [22], the successful implementation of KM and TQM practices helps managers to ensure that the level of production of employees is improved and that their efficiency is effective, which leads to the development and success of their companies. In other words, managers can make effective and precise decisions to manage their resources and ensure the motivation of their employees.

In general, employee effectiveness is necessary to successfully implement these two management philosophies, KM and TQM. However, are KM and TQM necessary to ensure the effectiveness of employees and improve their productivity? Several studies around the world have addressed this question by studying the effect of one or a few practices of KM and TQM on employees' performance. In this context, the relevance of this study is that it shows the effect of the most cited practices of KM and TQM in the literature on the effectiveness of employees in emerging economies. In order to reach these objectives, two hypotheses were considered. The first claims that TQM practices have a significant impact on employees' effectiveness, and the second consider that KM elements have a significant impact on employees' effectiveness. Thereby, the survey method was used, and collected data were analyzed using exploratory factor analysis, confirmatory factor analysis, and structural equation modeling. This method was developed to measure the effects of KM

elements and TQM practices and also to show the links between them in order to increase staff effectiveness and consequently maximize their performance.

This article is structured as follows: Section 1 presents the research's importance and purpose. Section 2 reviews TQM practices and KM elements and recaps some results from previous studies on the relationship between TQM, KM, and staff effectiveness. Section 3 exposes the proposed research model and its assumptions. Indeed, research methodology, survey instrument, and statistical analysis were outlined. Section 4 provides the empirical findings and analysis. After discussions, results, implications, and research limitations, the last part of the paper ends with conclusions.

## 2. Theoretical Background

Employees' performance is strongly linked to the overall productivity of companies through several factors. Most likely, these factors can mutually influence and interact with each other [23]. In this respect, several scientific publications have dealt with what can improve the collaborators' performance and how? In this respect, several scientific publications have dealt with what can improve the collaborators' performance and how? Hence, some researchers affirmed that the leadership style has a significant impact on employees' performance [1,24–26]. Likewise, organizational culture has a positive impact on employee performance [1,27]. Similarly, career development, management, and monitoring in the organization simultaneously have a significant effect on employee performance [28,29]. In addition, work discipline has a significant influence on employee performance and productivity [25,30,31]. Also, employee performance and morale are influenced by the work environment and workload [2,32], as well as employee work stress and employee conflict [29]. Moreover, employee effectiveness and productivity are highly dependent on occupational safety [30], work security [33], and job satisfaction [29,34]. Furthermore, most factors that best improve employees' performance are motivation [30,34,35], engagement and implication [3], compensation, and reward [4,33,36]. However, [37] it has also been proven that competence, digital transformation, and updating and improving skills have a positive and significant effect on employee performance.

From these claims in the literature, it is noticed that the majority of attributes that have an insightful impact on employees' effectiveness are included in KM and TQM practices. So, we will try in this section to summarize the main practices of both KM and TQM.

### 2.1. TQM Practices

TQM is a key strategy that leads organizations to improve overall effectiveness as well as performance to achieve an exceptional status [18]. It is a management approach that emerged to improve the quality of products and services and, through that, to strengthen organizational performance [38]. According to Kumar and Sharma [39], TQM is a process led by senior management to engage all employees in the continuous improvement of the performance of all activities. It offers guiding principles that emphasize the continuous improvement of products and services to meet or possibly exceed the requirements or expectations of the customers' organization [40]. The lack of commitment from top management in TQM programs has been one of the most important obstacles to the success of TQM [41]. This proves the importance of strengthening top management commitment reinforcement to TQM in order for it to succeed and add value to companies since, without it, unfortunately, nothing will happen. For Banuro et al. [42], TQM is a corporate philosophy in which all employees are involved in creating customer value by providing high-quality products/services. In fact, TQM practices have been implemented by companies interested in promoting their survival prospects by integrating quality and continuous improvement into their strategic priorities. Indeed, these practices serve to follow a set of management concepts and tools aimed at involving managers and employees in the achievement of continuous performance improvement [43]. Several authors have identified several success factors and practices of TQM [44]. According to Lehyani et al. [45], the six most important practices of TQM identified in the scientific literature are:

- Leadership: the phenomenon of leadership is described as any kind of process or act of influence that, in any way, causes people to do something [42]
- Customer focus: measures the degree to which the organization recognizes the needs of its customers and takes appropriate action, ensures their satisfaction, addresses their complaints, and provides any type of after-sales or recovery service [46]
- Human Resources Management (HRM): measures employee engagement, motivation, and satisfaction as well as maintaining a peaceful and enjoyable work environment that encourages quality improvement [47]
- Process management: organizational excellence is linked to good process management to ensure that processes are aligned with the organization's strategic objectives [48]
- Information analysis: measuring and analyzing information helps to assess the quality of processes and products/services as the most important method to verify continuous improvement, monitor processes, analyze and correct deviations from required standards [49]
- Continuous improvement: consists of improving outcomes and capabilities to produce better outcomes, improving strategic planning and high-level decision-making in the detailed execution of work elements in the workshop [39].

### 2.2. KM Elements

KM plays an important role in an organization's survival and competitive strength [50]. It is like access, evaluation, management, organization, filtering, and distribution of information in a way that would be useful to end users through a technology platform [51]. For Ali [52], KM is the process of capturing, distributing, and using knowledge effectively. It is a process that clearly establishes and articulates these linkages [53]. Nevertheless, according to Barley et al. [54], KM is a set of processes through which knowledge moves from differentiated states to integrated states to produce operational benefits. According to the work of Lehyani and Zouari [55], the five most cited elements of KM in the scientific literature are:

- Knowledge acquisition: occurs before learning is a process of reflection, allowing the individual to understand the acquired knowledge. It is also the main capital of a modern company that becomes a necessity, as well as the development of internal skills [56]
- Knowledge application: its purpose is to improve the effectiveness of an organization's management to know the weaknesses and strengthen the strengths to increase employee productivity in order to gain a competitive advantage [57,58]
- Knowledge creation: is the driver of sustained performance [59]. It means being more active and smarter at solving problems and overcoming obstacles which also means being more robust in an unstable and disruptive environment [60]
- Knowledge sharing/transfer: the activities of transferring or disseminating knowledge from one person, group or organization to another [61]. If individual knowledge is not shared with other individuals or groups, knowledge will have a limited impact on organizational effectiveness [54]
- Knowledge capitalization: it involves identifying, extracting, formalizing, preserving, and sustaining critical knowledge [62].

Other theoretical links are presented in Section 3 to introduce the strong relationship between the concepts of KM, TQM, and employee effectiveness.

### 3. Research Model and Proposed Hypotheses

Based on the work of Wickramasinghe [63], Durairatnam et al. [38], and Friedrich et al. [64], etc. the proposed research model of the relationship between TQM practices, KM elements, and employees' effectiveness is shown in Figure 1.

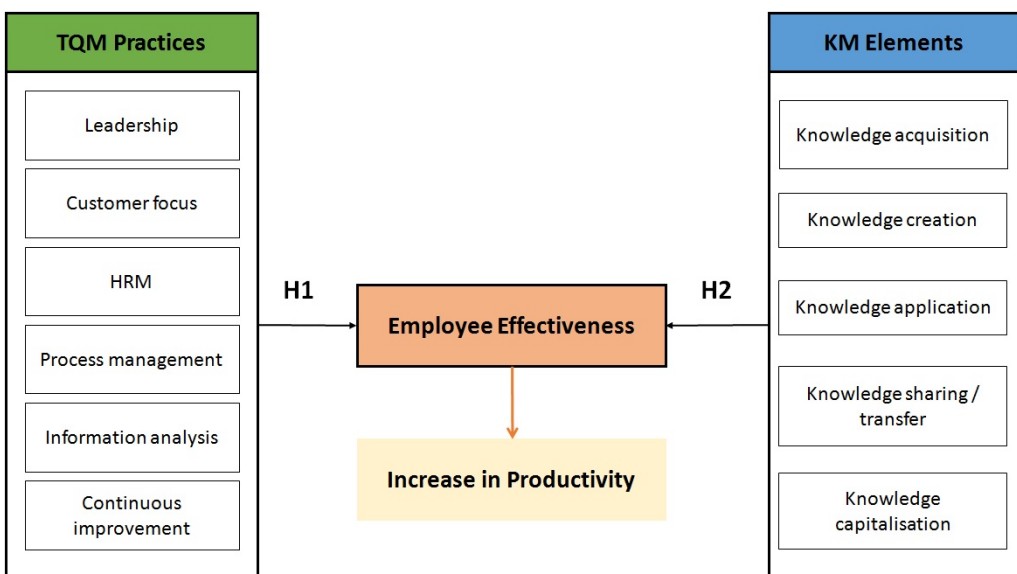

**Figure 1.** Proposed research model of the relationship between TQM practices, KM elements, and employee effectiveness.

According to Usrof and Elmorsey [18], employees' effectiveness and TQM are strategically and tactically important to gain competitive advantage through a variety of factors. TQM strives to disseminate fundamental skills and knowledge to all employees in order to improve quality and give them the confidence to achieve effective results from their work [38]. Besides, Mohammed et al. [65] have stated that the key factors of TQM have a direct effect on job performance. Depending to Sadikoglu and Zehir [66], employees do not feel fear or job insecurity because top management initiates open communication between them and supports their involvement and empowerment. Consequently, employees feel a sense of ownership, self-actualization, loyalty to the organization, pride in their work, and satisfaction from that work. However, the TQM application pushes managers to treat their employees like customers because they are people who are integrated into society and, at the same time, are part of the organization. As a result, employees can express themselves freely and have what they need [67].

TQM's successful implementation requires additional professional training, which is critical to its success and provides an opportunity to empower and motivate employees [68]. Likewise, Wickramasinghe [63] has claimed that TQM practice application influence employee satisfaction and their well-being through rewards, recognition, recruitment, and good HR planning. For Wang et al. [69], organizations that implement TQM can improve staff job satisfaction, which can influence organizational culture. In this vein, Obeidat et al. [70] have assured that enhancing the competitiveness of an organization is related to TQM because it improves the level of employee empowerment and leadership. As a result, HRM and TQM will have significant positive effects on the competitiveness of human resources [19]. Thus, the following hypothesis is proposed:

**Hypothesis 1 (H1):** *TQM practices have a significant impact on employees' effectiveness.*

Woolliscroft et al. [71] have proven that the core of KM's development is improving employee expertise. Indeed, Lendzion [72] has asserted that KM can strengthen the company's competitiveness through employees' knowledge and skills, hence their effectiveness. However, Masa'deh et al. [73] have confirmed that an appropriate KM infrastructure in organizations improves the interaction between employees and establishes a close relationship between them. In this context, effective training will augment employees' knowledge and their ability to learn, thus strengthening their loyalty to the firm, their motivation, and their work performance [48].

On the other hand, KM systems enable employees to effectively use the accumulated knowledge and expertise within the organization, leading to improved employee yield and performance [74]. Furthermore, Agrawal and Mukti [75] have found linkages between KM and employees' effectiveness, which appears in enhancing the key skills of employees and building the company's competitiveness. Latif [76] has thought that if staff are equipped with new knowledge, skills, and attitudes, organizations can ensure their survival and improve their ability to cope with market change. Based on the reviewed literature, the following hypothesis is suggested:

**Hypothesis 2 (H2):** *KM elements have a significant impact on employees' effectiveness.*

The approval of the proposed hypothesis and the improvement of employee effectiveness can be translated into a consistent increase in the productivity of the company, even in its global performance.

## 4. Research Methodology

This work is based on a survey designed to study KM and TQM impact on staff effectiveness in Tunisian Small and Medium Enterprises (SMEs). The survey has been divided into two parts. Part 1 contains questions related to the descriptive details of enterprises. Part 2 consists of open-ended questions that respondents can answer using a Likert scale ranging from 1 to 5, from very low impact to very high impact.

A cross-sectional survey methodology is used in this study, and the survey was distributed in two ways: direct contact with 125 companies and sending emails to approximately 3000 companies from several fields. With this strategy, the percentage of responses obtained by direct contact is around 80%, and by sending emails is about 3.5%. Hence, the total number of obtained responses was 206. However, questionnaire validation was carried out by academic and non-academic experts who checked the nature of the questions, their understanding, consistency, etc. The instrument had a Cronbach's Alpha of 0.986, which is considered reliable [77]. Indeed, survey questions were coded to be processed by "SPSS" software version 24. Then, the factor analysis method and the independence test were employed.

### 4.1. Exploratory Factor Analysis

For the identification of factors with eigenvalues of at least one and to capture the factor loads that are easier to interpret, an Exploratory Factor Analysis (EFA) was carried out through the use of the main component's extraction method with varimax rotation. In order to make the results more credible and reduce potential deviations from the missing variables, we added the size and type of enterprise variables as control variables in the equations. Besides, to determine the correlation of TQM factors and KM elements with each other and with staff effectiveness, we have performed a bivariate correlation analysis.

### 4.2. Confirmatory Factor Analysis and Structural Equation Modeling

In fact, Confirmatory Factor Analysis (CFA) was used to improve the resulting scales in EFA and to assess the convergent validity and the unidimensionality of measures. After confirmation of the validity and reliability of the latent variables, the model and the assumptions were tested using Structural Equation Modeling (SEM) through path analysis. Regarding the effects of significant correlations and the possibility of multicollinearity problems, they were reduced by applying SEM. Then, the model was assessed by examining the goodness-of-fit statistics indices: ratio of $\chi^2$ to the degree of freedom, Root Mean Square Error of Approximation (RMSEA), Parsimony Goodness of Fit Index (PGFI), Akaike's Information Criterion (CAIC), Parsimony Normed Fit Index (PNFI), and Comparative Fit Index (CFI).

## 5. Results of the Analysis

### 5.1. Sample Demographics

As a sample of Tunisian companies, we have contacted around 3000 companies randomly. Table 1 provides a descriptive summary of 206 participating companies and respondents.

**Table 1.** Summary of participated companies and respondents.

| Relevant Dimension | Profiles in Percentages | Relevant Dimension | Profiles in Percentages |
|---|---|---|---|
| Localisation | 37.4% Sfax | Activity field | 18.9% Service |
| | 17% Tunis | | 5.8% Petro-chemical industry |
| | 7.3% Nabeul | | 6.8% Electric and Electronic |
| | 4.4% Ariana | | 2.9% Training |
| | 4.9% Ben arous | | 1.9% Health |
| | 4.9% Sousse | | 7.8% Construction |
| | 4.4% Mahdia | | 5.8% Transport and distribution |
| | 3.4% Bizerte | | 17% Agri-Food |
| | 3.4% Monastir | | 5.3% Automotive |
| | 2.9% Gabes | | 13.6% Manufacturing |
| | 2.4% Medenine | | 5.8% Textile |
| | 2.4% Zaghouan | | 4.4% Materials |
| | 1% Beja | | 3.9% Telecommunication |
| | 1% Jendouba | Number of employees | 15.5% less than 9 |
| | 0.5% Gafsa | | 25.7% between 10 and 49 |
| | 0.5% Kairouan | | 26.7% between 50 and 249 |
| | 1.5% Manouba | | 32% More than 250 |
| | 0.5% Sidi bouzid | | 24.8% Quality Manager |
| | 0.5% Tataouine | | 17.5% Human Resources Manager |
| | 51.5% QMS | | 16% Logistics Manager |
| | 16.8% EMS | | 6.8% Production Responsible |
| | 9.9% OHSMS | Respondent function | 4.4% Purchasing and Supply Manager |
| | 5% FSMS | | 6.8% Sales and Marketing Manager |
| Certification | 3% SMS | | 7.3% Technical Manager |
| | 2% Others | | 2.9% Financial Officer |
| | 9.9% Not certified | | 3.4% Administrative Officer |
| | 2% In progress | | 10.2% General Manager |

### 5.2. Exploratory Factor Analysis Results

In order to confirm sampling adequacy, data homogeneity was measured prior to the Principal Component Analysis (PCA). Table 2 shows the data matrix for the analysis of TQM practices, KM elements, and employee effectiveness. From this table, it can be noted that the KMO value is greater than the value of 0.7 [66] and the value of the Barlett Test (BT) is less than 0.05. Therefore, it can be concluded that the data show suitability and homogeneity conducive to PCA application.

**Table 2.** Data matrix for the analysis of TQM practices, KM elements and employee effectiveness.

| Kaiser–Meyer–Olkin (KMO) Measure of Sampling Adequacy | | 0.927 |
|---|---|---|
| Bartlett's Test (BT) of Sphericity | Approx. Chi-Square | 2253.240 |
| | df | 136 |
| | Sig. | 0.000 |

In EFA, we have used independent and dependent variables (TQM practices, KM elements, and employee effectiveness) together. The final measurement instruments, which consist of six TQM practices, five elements of KM, and employee effectiveness, explain 67.58% of the total variance considering an eigenvalue of more than one, see Table 3. In this Table, the t-value represents the CFA coefficient test of each element.

**Table 3.** Rotated factor matrix of the TQM practices, KM elements, and employee effectiveness.

| | Variables | Component 1 | Factor Loadings Component 2 | Component 3 | Eigenvalue | Percentage Variance Explained by Factor | Percentage Total Variance Explained | t-Value |
|---|---|---|---|---|---|---|---|---|
| Employee Effectiveness | Employee expertise | - | 0.747 | - | | | | 11.42 |
| | Continuous employee learning | - | 0.648 | - | | | | 9.71 |
| | Employee involvement and commitment | - | 0.765 | - | 8.72 | 51.31 | 51.31 | 11.28 |
| | Integration/interaction with employees | - | 0.757 | - | | | | 11.28 |
| | Employee performance | - | 0.788 | - | | | | - |
| | Use of technological resources | - | 0.776 | - | | | | 11.05 |
| KM | knowledge acquisition | 0.690 | - | - | | | | - |
| | knowledge creation | 0.778 | - | - | | | | 11.87 |
| | knowledge application | 0.785 | - | - | 1.66 | 9.78 | 61.09 | 11.04 |
| | knowledge sharing/transfer | 0.773 | - | - | | | | 11.59 |
| | knowledge capitalization | 0.731 | - | - | | | | 11.75 |
| TQM | Leadership | - | - | 0.642 | | | | 11.25 |
| | Customer focus | - | - | 0.776 | | | | 8.29 |
| | HRM | - | - | 0.732 | 1.10 | 6.49 | 67.58 | 12.08 |
| | Process management | - | - | 0.725 | | | | 12.79 |
| | Information management | - | - | 0.659 | | | | 13.31 |
| | Continuous improvement | - | - | 0.767 | | | | - |

### 5.3. Tests for Reliability and Validity of the Constructs

For internal consistency, the reliability of the multiple-item measurement scale was measured via Cronbach's alpha. To ensure that the scale items measure the corresponding latent variables consistently and that they are free of measurement errors by using Cronbach's alpha, we have used reliability analyses [73]. However, if these items are removed from the analysis, there will be no significant increase in structural reliability. Nevertheless, the reliability analysis of TQM practices was 0.884, and the reliability analysis of KM elements was 0.899. In fact, Table 4 presents descriptive statistics, values of Cronbach's alpha, and Pearson correlations of the variables in our research model. Also, it presents Standard Deviation (SD), which is the degree of dispersion or the scatter of the data points relative to its mean. This table shows that the alpha values of all variables surpassed the 0.70 threshold. Therefore, according to Wang et al. [69], it can be concluded that it has excellent reliability.

The validity of the model was tested before the use of SEM. In addition, CFA was performed for the variables resulting from EFA to examine them in order to determine whether they measure our model elements. All TQM practices, KM elements, and employee effectiveness have statistically significant factor loads on their designated scales. Indeed, the measurement model was estimated without limiting the factor covariance matrix. However, considering the significance level of 0.001, the measurement model is statistically acceptable. Table 3 shows the retained measurement variables, the corresponding standardized factor loadings, and t values resulting from testing the items' coefficients in CFA.

Through a subjective assessment of the validity of the 17 dimensions (factors) of our work, we can grant content validity. Thus, the content validity of these factors is ensured by an exhaustive review of the literature. Construct validity assesses the degree of association between a single element and the scale. Factor saturation determines the unidimensionality of the metric used in this study. Consequently, we have kept the elements with a saturation factor of at least 0.50. As shown in Table 3, all factor loadings of TQM practices, KM elements, and employee effectiveness exceed the 0.50 threshold.

According to Tarí et al. [78], convergent validity should be more than 0.50. It is assessed using items loading, Composite Reliability (CR), and Average Variance Extracted (AVE). As shown in Table 5, the AVE for all constructs and CR are more than the recommended threshold of 0.5, which signifies the convergent validity of the model.

**Table 4.** Descriptive statistics, Cronbach's alpha, Standard Deviation (S.D), and bivariate correlations for the variables in the research model *.

| | Variables | 1 | 2 | 3 | 4 | 5 | 6 | 7 | 8 | 9 | 10 | 11 | 12 | 13 | 14 | 15 | 16 | 17 | Mean | S.D | Alpha Value |
|---|---|---|---|---|---|---|---|---|---|---|---|---|---|---|---|---|---|---|---|---|---|
| 1 | Employee expertise | 1.00 | | | | | | | | | | | | | | | | | 3.67 | 0.84 | |
| 2 | Continuous employee learning | 0.65 | 1.00 | | | | | | | | | | | | | | | | 3.51 | 0.86 | |
| 3 | Employee involvement and commitment | 0.59 | 0.64 | 1.00 | | | | | | | | | | | | | | | 3.58 | 0.84 | 0.896 |
| 4 | Integration/ interaction with employees | 0.57 | 0.61 | 0.64 | 1.00 | | | | | | | | | | | | | | 3.59 | 0.92 | |
| 5 | Employee performance | 0.41 | 0.64 | 0.61 | 0.65 | 1.00 | | | | | | | | | | | | | 3.57 | 0.83 | |
| 6 | Use of technological resources | 0.49 | 0.56 | 0.62 | 0.59 | 0.67 | 1.00 | | | | | | | | | | | | 3.53 | 0.84 | |
| 7 | knowledge acquisition | 0.50 | 0.47 | 0.45 | 0.41 | 0.45 | 0.41 | 1.00 | | | | | | | | | | | 3.65 | 0.81 | |
| 8 | knowledge creation | 0.48 | 0.47 | 0.43 | 0.47 | 0.46 | 0.42 | 0.65 | 1.00 | | | | | | | | | | 3.54 | 0.97 | |
| 9 | knowledge application | 0.37 | 0.44 | 0.39 | 0.37 | 0.45 | 0.35 | 0.58 | 0.70 | 1.00 | | | | | | | | | 3.68 | 0.92 | 0.899 |
| 10 | knowledge sharing/transfer | 0.44 | 0.43 | 0.49 | 0.38 | 0.42 | 0.35 | 0.57 | 0.65 | 0.65 | 1.00 | | | | | | | | 3.71 | 1.02 | |
| 11 | knowledge capitalization | 0.45 | 0.46 | 0.51 | 0.40 | 0.49 | 0.42 | 0.62 | 0.67 | 0.60 | 0.73 | 1.00 | | | | | | | 3.55 | 1.01 | |
| 12 | Leadership | 0.38 | 0.37 | 0.43 | 0.34 | 0.40 | 0.37 | 0.46 | 0.50 | 0.46 | 0.51 | 0.49 | 1.00 | | | | | | 3.85 | 0.90 | |
| 13 | Customer focus | 0.29 | 0.37 | 0.31 | 0.28 | 0.33 | 0.31 | 0.31 | 0.35 | 0.29 | 0.33 | 0.36 | 0.51 | 1.00 | | | | | 3.73 | 0.94 | |
| 14 | HRM | 0.41 | 0.37 | 0.43 | 0.41 | 0.43 | 0.33 | 0.43 | 0.48 | 0.46 | 0.54 | 0.53 | 0.58 | 0.52 | 1.00 | | | | 3.74 | 0.94 | 0.884 |
| 15 | Process management | 0.40 | 0.46 | 0.41 | 0.40 | 0.42 | 0.37 | 0.42 | 0.46 | 0.46 | 0.49 | 0.52 | 0.47 | 0.52 | 0.64 | 1.00 | | | 3.73 | 0.94 | |
| 16 | Information management | 0.45 | 0.42 | 0.52 | 0.46 | 0.39 | 0.34 | 0.46 | 0.59 | 0.50 | 0.66 | 0.58 | 0.56 | 0.37 | 0.64 | 0.62 | 1.00 | | 3.77 | 0.94 | |
| 17 | Continuous improvement | 0.43 | 0.44 | 0.45 | 0.36 | 0.39 | 0.39 | 0.50 | 0.60 | 0.61 | 0.59 | 0.57 | 0.60 | 0.39 | 0.55 | 0.70 | 0.67 | 1.00 | 3.95 | 0.93 | |

* N = 206, all correlations are significant at the $p < 0.001$.

**Table 5.** Evaluation of the measurement model.

|  | Variables | Composite Reliability (CR) > 0.70 | AVE > 0.50 |
|---|---|---|---|
| Employee Effectiveness | Employee expertise<br>Continuous employee learning<br>Employee involvement and commitment<br>Integration/ interaction with employees<br>Employee performance<br>Use of technological resources | 0.884 | 0.560 |
| KM | knowledge acquisition<br>knowledge creation<br>knowledge application<br>knowledge sharing/transfer<br>knowledge capitalization | 0.867 | 0.566 |
| TQM | Leadership<br>Customer focus<br>HRM<br>Process management<br>Information management<br>Continuous improvement | 0.864 | 0.516 |

The discriminant validity of the measurement model can be estimated by comparing the values with the square root of the AVE in a diagonal with the correlation among reflective constructs. These values should be greater than the correlation coefficients between the constructs (see Table 6).

**Table 6.** Fornell–Larcker's method.

|  | KM | TQM | EE |
|---|---|---|---|
| KM | (0.752) |  |  |
| TQM | 0.292 | (0.718) |  |
| EE | 0.348 | 0.275 | (0.748) |

Note: The square roots of the AVE are the values enclosed in parentheses (the diagonal values). The correlations between the constructs are the other values of the matrix. In order to verify discriminant validity, off-diagonal values must be less than diagonal values.

The results of the bivariate correlation show that the TQM and KM variables have positive and significant correlations that demonstrate the interdependence of TQM practices and KM elements. Because of the high standard errors of the estimated correlation coefficients, inferences based on the regression estimates can be difficult to determine because of the proportionately high multicollinearity between the independent variables.

As shown in Table 4, correlations between variables are less than 0.8, indicating that there is a strong positive correlation between them [73]. However, the Variance Inflation Factor (VIF) was measured to examine multicollinearity. VIF values for our data were less than three. This result indicated that multicollinearity does not exist in our data and does not have an undue effect on the least squares estimates [48]. Besides, the assumption of normality is satisfied following the normality tests of the variables adopted in our measurement model.

*5.4. The Structural Model Test Results*

SEM was used to test our model. Thus, Table 7 presents the results of the measurement model and the structural model as well as the recommended values of the fit indices for the satisfactory fit of a model to data. For validating the structural model, we have compared the goodness-of-fit statistics with the recommended values of the fit indices, and we have concluded that they did not exceed these values.

**Table 7.** Results of the measurement model and structural model.

| Goodness-of-Fit Statistics | Measurement Model for TQM, KM and EE | Structural Model | Recommended Values for Satisfactory Fit of a Model to Data |
|---|---|---|---|
| $\chi^2/\text{df}$ | 245.438/116 = 2.11 | 185.207/113 = 1.639 | <3 |
| Root mean square error of approximation (RMSEA) | 0.074 | 0.056 | <0.08 |
| Akaike's information criterion (CAIC) | 479.570 | 438.322 | <Saturated model and independence model |
| CAIC for saturated model | 968.165 | 968.165 | |
| CAIC for independent model | 2465.508 | 2465.508 | |
| Parsimony goodness-of-fit index (PGFI) | 0.668 | 0.668 | >0.5 |
| Parsimony normed fit index (PNFI) | 0.764 | 0.766 | >0.5 |
| Comparative fit index (CFI) | 0.942 | 0.968 | >0.9 |
| Goodness of Fit Index (GFI) | 0.881 | 0.905 | >0,8 |
| Adjusted Goodness of Fit Index (AGFI) | 0.843 | 0.871 | >0,8 |
| Root-mean-square residual (RMR) | 0.035 | 0.032 | ≤0.05 |

The relationship between TQM and KM with staff effectiveness is shown in Table 8. The results demonstrate that all assumptions are valid. It is noticed that the probability value (*p*-value), which is obtained from a statistical analysis, is calculated by its verification against the threshold value of the significance level in order to reject null assumptions.

**Table 8.** Construct Structural Model.

| Links in the Model | Hypotheses | | Standardized Parameter Estimates | | Results |
|---|---|---|---|---|---|
| | Number | Sign | Estimate | *p*-Value | |
| TQM → employee effectiveness | H1 | + | 0.20 | 0.019 | Supported |
| KM → employee effectiveness | H2 | + | 0.49 | 0.000 * | Supported |

Note: * $p < 0.1$.

*5.5. Discussion*

This study explored TQM and KM's impact on employee effectiveness in Tunisian companies. Hypothesis 1 confirmed that the effect of all TQM practices (leadership, customer focus, human resources management, process management, information analysis, and continuous improvement) on employee effectiveness was significant. This confirmation agrees with other studies that have treated this subject as [18–20]. This means that if companies excel in applying one of the TQM practices, they will obviously improve the level of employee effectiveness. Furthermore, the justice, wisdom, and temperance of leaders are positively associated with effectiveness and team performance [79]. Budur and Poturak [80] have proved that customer focus has a positive and significant impact on employee effectiveness and customer loyalty. Indeed, Al-Harazneh and Sila [81] have affirmed that firms' decisions can be affected by the employee's behavior and professionalism. So, they suggested the digitalization of their HRM that directly affects organizational and employee effectiveness. In addition, process management involves repeating and improving routines, noting that routines involve various procedures and skills that help employees to improve their functions [82]. Companies are encouraged to keep their information and analyze it to provide it to employees in a timely manner. In this case, clear information can be a source of excellence and development for certain employees, thus improving their effectiveness. Moreover, Oliveira et al. [46] have affirmed that continuous improvement is well connected with the organization's culture. It encourages employees' participation and leads them to

express their opinions and share knowledge between them, which strengthens employee effectiveness and performance.

Hypothesis 2 confirmed that the effect of all KM elements (knowledge acquisition, knowledge application, knowledge creation, knowledge sharing/transfer, and knowledge capitalization) on employee effectiveness was significant. Knowledge acquisition is a path to better understanding the knowledge acquired by employees. It is a necessity before the apprenticeship to develop the skills of the staff. Besides, knowledge application makes it possible to use the company's existing knowledge and to create new ones. The objective is to enrich the knowledge of the workers and to promote the effectiveness of the overall management of the organization. Furthermore, knowledge creation is based on the sharing of tacit and explicit employee knowledge within an organization. As a result, brains begin to produce, design and create more brilliant ideas. The success of this process generates the motivation and involvement of all employees. What is more, knowledge sharing/transfer minimizes processing and research time. This gain helps employees have the time they need to perform other tasks. It also strengthens relationships between them and builds trust and collaboration. In fact, preserving, securing, and capitalizing on an organization's knowledge is critical to sharing and using it with everyone when needed, which is a very important competitive advantage for these companies. This minimizes sterile and unnecessary activities and increases employee productivity and effectiveness. These results agree with several recent research like the article of Ogbu Edeh and Blessing [83], who concluded that KM enhances employee effectiveness quite apart from how jobs should be carried out. Moreover, KM has a significant consequence on employee and business efficiency [84]. Also, Syarifuddin et al. [85] state that KM has a significant effect on skills, attitudes, and employee performance. Likewise, organizational learning stimulates the positive effect of KM practices on employee performance [86].

## 6. Managerial Implications

This study examines the relationships between KM, TQM, and employee effectiveness by analyzing the links between KM elements and TQM practices on staff effectiveness. Based on previous research, this study found that KM and TQM are two key factors in employee motivation, engagement, and effectiveness. Another point is that the model proposed in our study can be used by managers to improve the level of effectiveness of their employees. They can determine the effect of TQM practices and KM elements on employee effectiveness by evaluating existing variables in their companies.

In fact, the results of our research lead managers to consider applying TQM and KM in their companies, noting that in the last decade, a lot of efforts have been made to integrate these two notions in Tunisian SMEs. With more attention and willingness on the part of management to implement TQM and KM effectively, companies can have a significant competitive advantage over companies that have not yet implemented them. Moreover, managers are advised to implement KM and TQM practices in their organizations to improve knowledge worker performance and the welfare of employees at work. These leaders need to commit more resources, time, and awareness to achieve a high level of employee effectiveness in order to achieve efficiency in the future. It should be noted that through the integration of the two concepts (KM and TQM), companies can improve the situation and working conditions of their employees, who become more productive, motivated, involved, and effective. Finally, if companies seek to strengthen their market share, enhance their brand image, optimize their resources, and support their capacity for innovation and development, they must give more attention to their employees by meeting their needs and making them feel that they are important and belonging. However, this can only be achieved through the application of TQM practices and KM elements.

One of the limitations of this work is the sample size. A higher number may lead to new results or other conclusions. Also, communication can be added to our research as a key factor of TQM practices. Besides, the obtained answers were from persons who did not have a clear idea of the theme of our research. That is, the answers could be more precise if

we were able to contact these persons face to face. In addition, most of the respondents were managers and senior employees (managers or directors) who have different opinions than workers. That is why we should have questioned workers in our investigation. Another limitation is that we did not cite practices or characteristics of employee effectiveness, and we considered it as a single criterion.

## 7. Conclusions

The main aim of this study was to investigate the effect of TQM practices and KM elements on staff effectiveness within the context of Tunisian companies. The findings in these firms showed a positive effect of TQM practices on staff effectiveness. This effect is adequate for leadership, customer focus, human resources management, process management, information analysis, and continuous improvement. From the results, it is remarkable that companies that care more about their management approaches can achieve higher levels of employee effectiveness than others. This can be achieved when managers focus on good evidence-based decision-making, quality programs, and training, ensuring a motivating and balanced work environment, etc.

Also, the study revealed that KM elements like knowledge acquisition, knowledge translation, knowledge creation, knowledge sharing/transfer, and knowledge capitalization improve staff effectiveness in Tunisian companies. Once the staff has acquired the necessary knowledge without putting in much effort or facing many obstacles, they will be motivated and comfortable to do their work. This will directly influence their performance and consequently their willingness to work. Thus, managers must ensure that all knowledge is shared among employees either through the implementation of a knowledge management system, experts, or even through the use of new technologies. This gives employees a sense of comfort, stability, and confidence that allows them to help their colleagues, have time to learn, innovate, and be involved in the company.

In fact, KM and TQM are two key strategies for companies to master their employees, their working conditions, and their knowledge. Thus, this study highlights the need for companies to cultivate the effectiveness of their staff by ensuring that TQM practices and KM elements are effective and well-implemented. Also, the study shows that through the mediation of TQM practices and KM elements, staff effectiveness can improve. Indeed, the study reveals that firms should focus on the proper management of their human resources by motivating their employees and renewing their skills and knowledge. That can be achieved by providing the necessary training and encouraging them to share and communicate their opinions. We also find that companies should encourage employees to apply their new ideas, focus on customer satisfaction and always try to retain them. As a result, firms will have greater economic value and will develop a sustainable competitive advantage. Finally, we encourage future researchers to carry out more studies on Tunisian companies in order to sensitize their leaders, develop their cultures and present Tunisia industrially. We also think that our study should be conducted in other countries to see whether, in other contexts with different cultural and management models, we are achieving the same results.

According to the results, Tunisian companies should equip themselves with knowledge management systems. Also, knowledge must be properly managed among employees to not create enmity and jealousy between them. Indeed, managers must set an example for their employees by sharing their knowledge and by carefully following all quality procedures and instructions issued by the company. Finally, managers should not only think about what they have paid but also about what they will benefit from. Furthermore, this study has some limitations; the main one is the lower number of participating companies in addition to the employment degree of the respondents, which can influence the comprehension and the accuracy of the answers. Furthermore, collected data comes from several industrial fields; this may include disagreement or misunderstanding and dispersion in their application level of practice of KM and TQM. Finally, the study is relatively general and did not analyze the maturity level of participating enterprises. As for further research

directions, we plan to investigate the effect of KM and TQM practices on facets of supply chain performance and to study how this might be impacted by staff effectiveness.

**Author Contributions:** Conceptualization, F.L. and A.Z.; methodology, F.L., A.Z. and A.G.; formal analysis, F.L., A.Z. and A.G.; data collection, F.L. and A.Z.; software, F.L. and A.G.; validation, A.Z., M.T., A.G. and J.C.S.; writing—original draft preparation, F.L. and A.Z.; writing—review and editing, F.L., A.Z., A.G., M.T. and J.C.S.; supervision, A.Z., M.T., A.G. and J.C.S. All authors have read and agreed to the published version of the manuscript.

**Funding:** This research received no external funding.

**Institutional Review Board Statement:** Not applicable.

**Informed Consent Statement:** Not applicable.

**Data Availability Statement:** All data are available upon request.

**Acknowledgments:** Authors express their gratitude and acknowledgement to all respondents to their survey.

**Conflicts of Interest:** The authors declare no conflict of interest.

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
