# Peer review of "Knowledge Management and Total Quality Management Impact on Employee Effectiveness in Emerging Industries: Case of Tunisian Small and Medium Enterprises"

_sustainability, doi:10.3390/su15053872_

Round 1

Reviewer 1 Report

Dear authors,

the contribution deals with an extremely interesting topic. The .The title of the article in my opinion should be shorter. From my point of view, I would recommend writing more about how research can fill the gap in the literature. I would improve the chapter on literature review. In the introduction, you could go into more general detail on the topic you are dealing with, also with reference to previous studies. I would develop the implications for theory, practice and policy makers in one chapter. The text is understandable and clear, but I would recommend reviewing the English language more. I recommend adding the sources to the figures in the text. Increase your references and include the most recent ones too. Combine the discussion chapter with the results chapter and delve into a little more detail.

Best regards

Reviewer 2 Report

Presented article with Title ”Knowledge Management and Total Quality Management Impact on Employee Effectiveness in emerging industries: Case of  Tunisian Small and Medium Enterprises ” is writing on 15 pages with 1 figures, 8 tables and 41 references. The paper is written clearly, but it has a number of shortcomings. The structure is very clear (introduction, Theoretical background, Research, methodology, results, discusion and conclusion), especially part of the results.

Suggestions:

-          Theoretical background need more source oriented to topic of article.

-          I recommend the authors to pay more attention to the chosen hypotheses.

-          Conclusions require renumbering and redrafting. They are too short and inconsistent.

-          Conclusions, please describe your future directions.

The quality and information provided in some tables needs correcting. All the specific comments can be followed in reviewed copy of the manuscript.     

I recomend this paper publish in journal after minor revision.

Reviewer 3 Report

Please see the comment file.

Reviewer 4 Report

Dear author(s),

Thank you for the opportunity to review this paper.  I agree that this is an important and pertinent topic. Although the idea is a good one, unfortunately, the way in which the study is operationalized holds back its potential contribution.  There are a few areas where I would encourage the authors to give further thought, as follows:

INTRODUCTION

The introduction should clearly illustrate (1) what we know (the key theoretical perspectives and empirical findings) and what do we not know (major, unaddressed puzzle, controversy, or paradox does the study addresses, or why it needs to be addressed and why this matters). And, (2) what will we learn from the study and how does the study fundamentally change, challenge, or advance scholars’ understanding. Much sharper problematization is required so that the introduction draws the reader into the paper. The introduction therefore needs to do a better job in setting the stage for the articulation of the theoretical contributions of the study. At the end of the introduction, we should have a clear idea of what the paper is about (i.e. its motivation, the gap in understanding that the paper is trying to address and summary of theoretical contributions).With references of 2022- 2020-2021.

Paragraph 1, with no references, explaining the context of the research.

Paragraph 2, with references, explaining very generally what we know about the topic introduced in Paragraph 1.

Paragraph 3 explaining what we need to find out.

Paragraph 4 explaining briefly what this paper   will do to find out, method etc.

Paragraph 5, with no references, explaining the structure of this paper.

LITERATURE REVIEW

  • Theoretical literature has not been considered and reviewed. It’s better to observe the connection between the contents. Try to explain everything except the topics in order to establish the necessary coherence.
  • Theoretical Development: The literature review must engage in the constructs of your analytical framing in a meaningful way. The literature review section could be improved by being more analytical. In other words, building on the existing literature to highlight what is missing and what is yet to be done and in so doing outline the theoretical puzzles or debates to which this work contributes. I have concerns related to theoretical development, and note the need for a more rigorous critique of the literature to help deepen the theoretical underpinnings of the study.

DISCUSSION

·       You need a discussion section. The discussion challenges your findings and determines the degree of compatibility with previous research.

·       The discussion section needs to highlight what is new in your findings and what we can learn from a study conducted in this interesting and understudied context. Whilst the introduction sets the stage for the study by justifying the relevance of the study, the discussion is the most important section as it is in the discussion that it is all brought together, and the authors illustrates how and why the study findings advance the literature. Therefore, the discussion needs to illustrate the new insights—the contributions—in a clear and compelling manner. In other words, illustrate what we know now that we did not know before or, in effect, to clearly illustrate the contribution of the study to the different bodies of literature. Furthermore, what are the future research directions based on this new framework?  

·       Theoretical Contributions: Addressing all the points mentioned above will lead to a more in-depth presentation of your data which has a clearer theoretical contribution. What is the theoretical contributions?  

CONCLUSION

·       The authors need to draw substantive conclusions from their results, and suggest, develop recommendations for further research.

·       What are the limitations of this research and how can it be solved by other researchers?

·       Use the following articles to improve the text of the article.

ü  Tajpour, M., Salamzadeh, A., Ramadani, V., Palalić, R., & Hosseini, E. (2022). Knowledge sharing and achieving competitive advantage in international environments: The case of Iranian digital start-ups. In International Entrepreneurship in Emerging Markets (pp. 206-224). Routledge.

ü  Tajpour, M., Hosseini, E., Mohammadi, M., & Bahman-Zangi, B. (2022). The effect of knowledge management on the sustainability of technology-driven businesses in emerging markets: The mediating role of social media. Sustainability, 14(14), 8602.

Best of luck with the further development of the paper.

Round 2

Reviewer 3 Report

The amendments are acceptable.

Reviewer 4 Report

Dear author(s), 

Thank you for the opportunity to review this paper.  I agree that this is an important and pertinent topic. Although the idea is a good one, unfortunately, the way in which the study is operationalized holds back its potential contribution.  There are a few areas where I would encourage the authors to give further thought, as follows:

The literature review must engage in the constructs of your analytical framing in a meaningful way. The literature review section could be improved by being more analytical. In other words, building on the existing literature to highlight what is missing and what is yet to be done and in so doing outline the theoretical puzzles or debates to which this work contributes. I have concerns related to theoretical development, and note the need for a more rigorous critique of the literature to help deepen the theoretical underpinnings of the study.

The conclusion part is more suggestion, it is better to write the main result of the research.

What are the limitations of this research and how can it be solved by other researchers?

Please attach the questionnaire.

Use references 2022-2023 in the literature and introduction section.

Tajpour, M., Hosseini, E., Mohammadi, M., & Bahman-Zangi, B. (2022). The effect of knowledge management on the sustainability of technology-driven businesses in emerging markets: The mediating role of social media. Sustainability, 14(14), 8602.
